# Oxidative Stress as a Potential Underlying Cause of Minimal and Mild Endometriosis-Related Infertility

**DOI:** 10.3390/ijms24043809

**Published:** 2023-02-14

**Authors:** Gabija Didziokaite, Gabija Biliute, Jogaile Gudaite, Violeta Kvedariene

**Affiliations:** 1Faculty of Medicine, Vilnius University, 03101 Vilnius, Lithuania; 2Department of Pathology, Institute of Biomedical Sciences, Faculty of Medicine, Vilnius University, 03101 Vilnius, Lithuania; 3Clinic of Chest Diseases, Allergology and Immunology, Institute of Clinical Medicine, Faculty of Medicine, Vilnius University, 03101 Vilnius, Lithuania

**Keywords:** oxidative stress, reactive oxygen species, infertility, unexplained infertility, endometriosis, mild and minimal endometriosis

## Abstract

Oxidative stress (OS) plays an important role in a variety of physiological and pathological processes of the female reproductive system. In recent years the relationship between OS and endometriosis has been of particular interest, and a theory has been suggested that OS may be a cause of endometriosis development. While the link between endometriosis and infertility is well established, minimal or mild stages of endometriosis are not considered to cause infertility. Increasing evidence of OS as a leading agent in the development of endometriosis has brought up a theory of minimal/mild endometriosis itself being one of the manifestations of high OS rather than a separate disease which directly causes infertility. Moreover, further development of the disease is thought to contribute to an increased production of reactive oxygen species (ROS) thus leading to the progression of endometriosis itself as well as to other pathological processes in the female reproductive system. Therefore, in case of minimal or mild endometriosis, the less invasive treatment could be offered in order to stop the ongoing cycle of endometriosis-reinforced excess ROS production and to reduce their harmful effects. In this article the existing connection between OS, endometriosis, and infertility is explored.

## 1. Introduction

Oxidative stress (OS) is defined as a state of imbalance between the production of reactive oxygen species (ROS) and the intrinsic antioxidant mechanisms [1]. Most ROS are continuously produced in the mitochondria when electrons leak from the electron transport chain [2]. It is estimated that around 2% of all oxygen used in the energy synthesis are diverted into ROS production [2]. Superoxide anion (O_2_−•), hydroxyl (•OH), and hydrogen peroxide (H_2_O_2_) are the three major types of ROS. Although hydrogen peroxide is not a free radical, its capability to cross biological membranes and convert into hydroxyl radical (•OH) is considered to have a deleterious effect on cells [2]. The oxygen radical species are unstable and highly reactive, thus in order to stabilize they need to acquire electrons from nearby molecules (such as carbohydrates, nucleic acids, lipids, or proteins) potentially causing lipid peroxidation, DNA, and protein damage [3]. Under normal conditions, potentially toxic chain reactions caused by ROS can be terminated by antioxidants, either by enzymes such as glutathione peroxidase, catalase, and superoxide dismutase or by non-enzymatic ones such as vitamins C and E [1]. However, when the antioxidant systems are impaired or overproduction of ROS occurs, the state of imbalance (better known as oxidative stress) is created and redox reactions happen in an uncontrolled manner leading to pathological change [1]. 

Oxidative stress plays an important part in a variety of different processes of the female reproductive system and fertility. Free radicals partake in several physiological processes in female fertility, including folliculogenesis, oocyte maturation, hormone signaling, tubal function, ovarian steroidogenesis, cyclical endometrial changes, germ cell function, feto-placental development, etc. when present in controlled amounts [1,2]. However, ROS have the potential to seriously harm cell structures when they reach pathological levels [1]. ROS can affect the microenvironments associated with follicular, hydrosalpingeal, and peritoneal fluid. Therefore, OS can have a direct impact on oocyte quality, activation, implantation, early embryonic development, and other processes related to the female reproductive system leading to OS-related female infertility [4].

In recent years, there has been growing evidence supporting the connection between chronic inflammation and oxidative stress, as triggering of inflammation leads to increased ROS generation which in turn enhances the pro-inflammatory gene expression [3]. The relationship between oxidative stress and endometriosis has been of particular interest.

Endometriosis is a chronic pelvic inflammatory disease defined by ectopic implantation and growth of endometrial tissue most often in the peritoneum, ovaries, and rectovaginal septum which can either be asymptomatic or cause chronic pelvic pain [3,5]. Endometriosis is a highly prevalent disease as it is present among 10–15% of women of reproductive age [5,6,7]. While it is a benign diagnosis, an estimated 30–50% of women with endometriosis are infertile and, according to different sources, up to 90% of infertile women are diagnosed with endometriosis [5,8,9,10]. Moreover, available treatment options tend to target symptom relief with surgical procedures often being unavoidable for refractory cases of infertility [11].

The link between endometriosis and infertility is well established, and if endometriosis is diagnosed, it is often considered to be the cause of infertility [7]. However, when minimal or mild stages of endometriosis are diagnosed, endometriosis is considered not to be the main cause of infertility [12]. If no secondary organic disruptions or additional pathologies associated with infertility are diagnosed, the diagnosis of unexplained infertility is usually established. This diagnosis does not allow couples to take targeted action to conceive and leaves them with the sole option of using artificial reproductive technologies. In this article, the existing connection between oxidative stress, endometriosis, and infertility is explored. 

## 2. Discussion

### 2.1. OS Association with Infertility

There are more than a few identified mechanisms through which OS may affect female fertility as ROS affects multiple physiological and pathological activities in the ovaries as well as in the peritoneal environment. Figure 1. The major sources of ROS are cytokines, macrophages, and leukocytes present in the follicular fluid microenvironment [2]. A specific amount of ROS is necessary as ROS are considered critical inducers of ovulation [2]. However, one of the ways female fertility might be affected by OS is by ROS effect on female germ cells. OS appears to be the main mechanism by which the postovulatory oocyte loses its developmental competence after ovulation [1]. A complex series of events, all driven by an increase in oxidative stress, makes postovulatory oocytes enter apoptosis and lose their functionality [1]. One typical sign of postovulatory oocyte aging, for instance, is zona pellucida induration, which can be triggered by exposure to OS [1]. Ovoperoxidase, which is found in the cortical granules that are discharged from the surface of the oocyte during an exocytotic process, is in turn powered by OS and facilitates oocyte aging [1]. Moreover, when ROS are produced in excess they can also damage oocyte’s DNA which may result in improper fertilization [1]. Studies have indicated that psychological stress and aging might cause oxidative imbalance [13]. Various studies have reported that an age-induced increase in oxidative stress is also related to the age-dependant increase in aneuploidy as OS was found to be involved in the non-dysjunction of chromosomes that characterize oocytes that have aged in vivo [1,13,14,15]. 

Furthermore, oxidative stress may have an effect on the granulosa cells’ ability to generate steroid hormones such as follicle-stimulating hormone (FSH) and estradiol (E2), which might affect the quality of oocytes [16]. The poor response to FSH and disturbed steroidogenic activity in older women may be connected to the increase in OS in the granulosa cells, which is associated with a decrease in the expression of the follicle-stimulating hormone receptor (FSHR) and a dysregulation of the FSHR signaling pathway. Women with endometriosis and PCOS have a lower antioxidant production capacity which may contribute to abnormal follicular development and infertility [16]. 

Some articles conclude that increased levels of stress hormones and decreased antioxidant activity augment the risk and prolong the duration of infertility [13]. The production of ROS beyond the physiological range (>80 ng/oocyte) is increased by stress-related pathological elevation of glucocorticoid levels. Excess ROS may lead to cell cycle arrest and apoptosis in follicular oocytes. They may affect not only ovarian but also uterine function, resulting in decreased fertilization and pregnancy rates [17].

Numerous studies have shown that PCOS patients also have increased OS. Patients with PCOS reported greater total oxidant status, higher serum prolidase activity, and higher OS indices, the proportion of oxidants to total antioxidant status. The mitochondrial malfunction in PCOS patients is explained by a reduction in mitochondrial O2 consumption and GSH levels as well as an increase in ROS generation. Increased levels of ROS produced by mononuclear cells during physiological hyperglycemia trigger the release of tumor necrosis factor (TNF) and increase the levels of a pro-inflammatory transcription factor, nuclear factor-kappa B (NF-κB). As a result, levels of TNF, a recognized modulator of insulin resistance, rise even more. The resulting OS induces prolonged anovulation, aberrant ovarian extracellular remodeling, cyst development, and an inflammatory environment that worsens insulin resistance, all of which contribute to infertility [2,18].

The association between higher levels of nitric oxide (NO) and nitric oxide synthase (NOS) with endometriosis and infertility has also been reported [19]. NO is a molecule of high physiological and pathological importance, whereas NOS is a family of enzymes catalyzing the production of NO [20]. NO is considered to be an essential molecule for normal reproductive biological processes such as sustaining pregnancy at physiological levels [19]. However, higher levels of NO have been reported to have harmful effects on sperm motility, toxicity to embryos, and inhibition of implantation [19]. It has been demonstrated that as the amount of NO in follicular fluid increases, the quality and rate of cleavage of the embryos are both reduced. Additionally, it has been noted that infertile women with tubal or peritoneal factor infertility had higher blood NO concentrations. Lower pregnancy rates are linked to follicular fluid NO concentrations that exceed the physiological limit, which can result in failed implantation. Studies conducted in vitro revealed that NO may even cause the embryo cells to undergo uncontrolled apoptosis and fragmentation [20]. 

Infections and ensuing oxidative stress are important factors in female reproductive health. For example, an opportunistic bacterium called *Enterococcus faecalis* is frequently discovered in the endometrium of patients with chronic endometritis. Superoxide-producing *E. faecalis* OG1RF infection of endometrial epithelial cells causes the production of inflammatory cytokines, encourages apoptosis, and suppresses the expression of receptivity indicators. Extracellular superoxide is a virulence factor of *E. faecalis*-induced endometritis that might result in decreased receptivity of endometrial epithelial cells and potentially lead to infertility [21]. 

Diminished ovarian reserve (DOR) seems to be associated with oxidative stress as well. DOR describes a decline in the number of oocytes in the ovary, which results in reduced female fertility and abnormalities of the reproductive endocrine system. OS can also cause ovarian endocrine dysfunction and follicular atresia, which is a major factor in the decline of fertility among patients with DOR [22].

DOR is a commonly encountered problem in reproductive medicine. While the levels of OS markers in follicular fluid are closely related to the growth, development, and maturation of the oocyte, it has been observed that changes in the DOR-associated metabolites in the follicular fluid may indicate the quality of oocytes. DOR-associated metabolites are a class of oxidative metabolites known as oxylipins, or lipid mediators, which are formed by the autooxidation of polyunsaturated fatty acids such as arachidonic acid, linoleic acid, alpha-linolenic acid, docosahexaenoic acid, and eicosapentaenoic acid as well as by the enzymes cyclooxygenase (COX), lipoxygenase (LOX), and cytochrome P450 (CYP). Nearly all physiological processes in the body involve the signal-transduction molecules oxylipins, which also play a crucial regulatory role in the organism’s daily activities such as immunological defense, oxidative stress, inflammatory response, and endocrine control [23,24].

Recent studies suggested that there were substantial changes in oxidative lipid metabolites between DOR patients and patients with normal ovarian reserve. The results of the study revealed 15 differentially expressed oxylipin metabolites, associated with ovarian reserve function. These fifteen oxylipin metabolites (±20-HDoHE, ±5-iso PGF2α-VI, 12S-HHTrE, 15-deoxy-Δ12,14-PGJ2, 1a,1b-dihomo PGE2, 1a,1b-dihomo PGF2α, 20-COOH-AA, 20-HETE, 8S,15S-DiHETE, PGA2, PGD2, PGE1, PGF1α, PGF2α, and PGJ2) were found to be of lower concentrations in the follicular fluid of DOR patients than of those in the normal ovarian reserve group. Oxylipin metabolism disorders and the function of ovarian reserve function were found to be closely related by metabolomic analysis of follicular fluid. While arachidonic acid is known to be involved in the regulation of oocyte development and maturation, it was observed that differentially oxidized lipid metabolites were mainly concentrated in the arachidonic acid (AA) metabolic pathway, therefore its complex changes might be closely related to impaired follicular development, leading to decreased fertility in DOR patients [24].

### 2.2. OS Association with Endometriosis

Endometriosis is considered to be one of the most common infertility-associated diseases. The first theories of endometriosis development were introduced almost 100 years ago. To this day, its etiology is still debatable and is usually based on one of the three main theories: retrograde menstruation and implantation theory by Sampson, coelomic metaplasia theory by Mayer, and the theory of induction [5]. However, for more than twenty years researchers have paid attention to the possible oxidative stress association with endometriosis development [5]. A number of oxidative stress biomarkers collected from different sites such as serum, peritoneal fluid, follicular fluid, ovarian cortex, eutopic, and ectopic endometrial tissues of women diagnosed with endometriosis were examined in different studies [5]. Based on the tendencies found, endometriosis was then characterized as the inflammatory process leading to the overproduction of inflammatory mediators due to oxidative stress [9]. When researching the exact role of oxidative stress in the pathophysiological mechanisms of endometriosis development, it was discovered that oxidative stress may be involved in multiple aspects of the disease and its progression. Figure 2.

It has been suggested that when endometrial cells get to the peritoneal cavity during retrograde menstruation, the immune system is activated and excess ROS are produced by activated macrophages, apoptotic endometrium cells, and erythrocytes [2,25]. Several studies have indicated that endometriosis patients have dysregulated immune systems; therefore, retrograde menstrual tissue has a higher chance of survival in the peritoneal environment [26]. In the case of menstrual reflux, macrophages are recruited to phagocytize apoptotic cells and release proinflammatory cytokines in the peritoneal cavity, which dilates and increases the permeability of the blood vessels thus enabling leukocyte extravasation [25]. High levels of malondialdehyde (MDA), pro-inflammatory cytokines (IL-6, TNF-α, and IL-1β), angiogenic factors (IL-8 and VEGF), monocyte chemoattractant protein-1 (MCP-1), and oxidized LDL (ox-LDL) were detected in the peritoneal fluid of endometriosis patients [2]. Pro-inflammatory and chemotactic cytokines are important in the recruitment and activation of phagocytic cells, which are the main producers of ROS [2]. Macrophages also degrade erythrocytes and release prooxidant and proinflammatory factors, such as hemoglobin and its highly toxic by-products heme and iron into the peritoneal environment during this process [5,25,27]. Hemolysis, together with a defective or overloaded peritoneal iron disposal system, leads to iron overload in the peritoneal environment [5,28]. As higher levels of iron, ferritin, and hemoglobin have been found in the peritoneal fluid of women affected by endometriosis, the role of altered iron metabolism in the development of endometriosis was further analyzed [5]. It was established that when iron is released into the peritoneal cavity, it can act as a catalyst in the Fenton reaction and generate a wide range of deleterious ROS including highly toxic hydroxyl free radicals, which can induce oxidative injury to cells [5,19,25]. Therefore, the literature suggests that iron-induced oxidative stress plays a key role and sets off the cascade of events that leads to the development and progression of endometriosis [5,19]. 

There are several mechanisms identified which support the theory that endometriosis, which may be caused by OS itself, may contribute to an even more intensive ROS production and therefore lead to the progression of this disease. For example, the ROS-producing enzyme xanthine oxidase is expressed in greater quantities among women diagnosed with endometriosis [2]. Moreover, endometrial toxins such as dioxin may cause an inflammatory-like process in the endometrium. This can worsen the expansion of endometriosis. The levels of paraoxonase-1, an enzyme that inhibits an oxidative change of low-density lipoprotein (LDL) cholesterol, are decreased in patients with advanced stages of the disease, whereas oxidized LDL levels are increased in the peritoneal fluid. This imbalance promotes the further spread of endometriosis [29].

As mentioned above, ROS production increases the levels of nuclear factor-kappa B (NF-κB) in peritoneal macrophages, which play a role in the immune and inflammatory response in endometriotic cells and lead to the increase in levels of cytokines, chemokines, adhesion molecules, growth, and angiogenic factors, in these cells [5,19,30]. The increased concentrations of cytokines, activated macrophages, and growth factors in the peritoneal fluid of endometriosis patients are also toxic to embryo survival and sperm function [19]. In some studies, endometriosis is also reported as an angiogenesis-dependent disease. OS can enhance vascular endothelial growth factor (VEGF) production in the endometrium, which itself plays an essential role in promoting angiogenesis. Therefore, OS can also promote the growth of endometrial implants. This effect is exerted in part by glycodylin, which is a glycoprotein whose expression is stimulated by oxidative stress as well [19]. Glycodylin increases VEGF expression in ectopic endometrial cells and stimulates their proliferation [19]. Additionally, retrograde menstruation-mediated hyperactivated oxidative stress may lead to stimulation of the extracellular signal-regulated kinase (ERK) and PI3K/AKT/mTOR signaling pathways and promote adhesion, angiogenesis, and proliferation of endometriotic lesions as well as subsequent endometriosis progression [26,31]. 

Previously mentioned NO and NOS levels were also reported to be higher in the endometrium of endometriosis patients, and NO levels were found to increase in the peritoneal fluid of infertile endometriosis patients [19]. Abnormal stimulation of endothelial NOS to release NO may be caused by different cytokines secreted from immune cells, endometrial cells, or macrophages. According to the hypothesis, stimulation of macrophages in patients with endometriosis may be a result of IL-10, which is augmented within earlier stages of endometriosis. Additionally, the expression of endothelial NOS in the endometrium also increases during the menstrual cycle of endometriosis patients. This abnormal stimulation of NO production in endometriosis patients results in excess levels of NO, which may lead to the inhibition of implantation [19].

While normally mesothelium is non-adhesive, ROS can also create adhesion sites on the peritoneal mesothelium of endometriosis patients that serve for implantation and progression of endometriosis [5,25]. Haemoglobin has also been identified as potentially harmful to mesothelium, leading to adhesion formation [5]. Adhesions themselves create blockades and may physically disturb the movement of oocytes and sperm. 

Different biomarkers associated with oxidative stress have been analyzed recently; however, some of the results were conflicting. Amreen et al. attempted to correlate and understand the relationship between oxidative stress biomarker levels and antioxidant levels measured in patients and its relation with the severity of endometriosis [32]. A statistically significant association between the increase in oxidative stress and the severity of endometriosis was observed. The study concluded that the median activity of lipid peroxide was highest in the severe stage of endometriosis (staged by the revised American Society for Reproductive Medicine scoring) in both blood and peritoneal fluid samples, whereas the median activity of antioxidants superoxide dismutase (SOD) and glutathione peroxidase was the lowest in the severe stage of endometriosis. Moreover, higher levels of lipid peroxide were also noticed in patients with dysmenorrhea. High levels of lipid peroxide in peritoneal fluid but not in blood had a significant correlation with the size of endometriotic lesions, and the tendency of higher lipid peroxide levels in the peritoneal fluid was observed among patients with chronic pelvic pain, supporting the opinion that a key inflammatory environment associated with endometriosis is found in peritoneal fluid [32]. It was suggested that oxidative stress biomarkers in serum may reflect oxidative status which is due to other causes, in addition to endometriosis, while markers, localized in the peritoneal fluid, may give a more accurate result [32].

There is a clear clinical need for new therapeutic approaches in the cases of minimal and mild endometriosis to curtail its recurrence and progression. As oxidative stress may be one of its causative agents, antioxidative treatment could be considered. Murine models have been used in several studies to examine ROS as potential therapeutic targets for the treatment of endometriosis. Recent studies describe a variety of antioxidants acting through different pathways to decrease levels of ROS, such as naringenin, resveratrol, curcumin, N-acetylcysteine [27]. Though there is evidence of increased ROS and decreased antioxidant levels in the pelvic and follicular environments of women with minimal and mild endometriosis, supplementation with immunomodulators that could reduce inflammation and oxidative stress such as pentoxifylline, or antioxidants like vitamin C and E, are found not to increase the likelihood of conception [27]. However, we believe that with the help of novel therapeutic approaches that focus on ROS imbalance and compromised mitochondrial function, our understanding will undoubtedly advance in the near future, resulting in more effective and safe treatments for endometriosis patients.

## 3. Conclusions

Oxidative stress plays a large role in the variety of different physiological and pathological processes of the female reproductive system and fertility. Endometriosis is one of the most common infertility-associated diseases and is suggested to be one of the causes of increased oxidative stress. However, there is evidence that increased ROS production prompted by other factors could also be the cause of endometriosis development as well as co-existing oxidative-stress-related infertility. This theory could explain the underlying OS-related cause of infertility in cases when only minimal or mild endometriosis is diagnosed and no adhesions or blockades of fallopian tubes are visualized during laparoscopy. Therefore, when minimal or mild endometriosis is diagnosed without any additional pathologies in the case of female infertility, OS could be considered the main reason for infertility, and endometriosis is one of the consequences of excess ROS due to increased OS. Moreover, while endometriosis itself may be caused by OS, there is evidence suggesting that when this disease develops it can also contribute to an even more increased production of ROS leading to the progression of the disease. Therefore, instead of establishing a diagnosis of unexplained infertility and suggesting artificial reproductive technologies, less invasive treatment might be offered for these patients in order to reduce the harmful effects of ROS and stop the ongoing cycle of endometriosis-reinforced excess ROS production. For example, some research suggests that antioxidant treatment may have possible beneficial effects.

Further investigation of OS effects on the peritoneal fluid environment could help acquire new insights into OS-related infertility and endometriosis development, leading to the development of novel diagnostic and therapeutic remedies for both pathologies. Additional research is necessary in order to prove oxidative stress as the main infertility-causing factor in the case of minimal and mild endometriosis, while the validation of putative biomarkers for OS is crucial to make progress in the field.

## Figures and Tables

**Figure 1 ijms-24-03809-f001:**
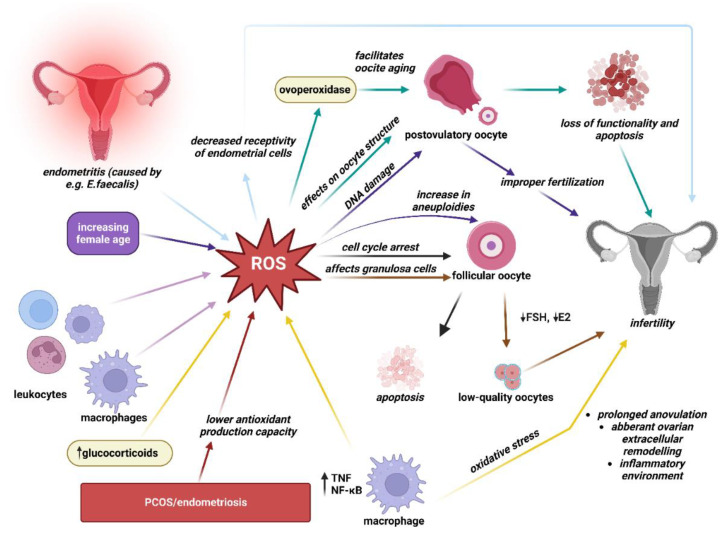
Identified mechanisms of oxidative stress-related infertility. The production of reactive oxygen species (ROS) during various processes (such as aging, uterine infections, etc.) and by various sources (such as leukocytes, macrophages, etc.) are illustrated via multi-colored arrows. Arrows of respective colors in turn represent the mechanisms by which ROSs may lead to infertility, e.g., *E. faecalis* endometritis may lead to an increase in ROS production, which in turn have an effect of decreasing receptivity of endometrial cells thus potentially leading to impaired fertility. (This figure was created with BioRender.com).

**Figure 2 ijms-24-03809-f002:**
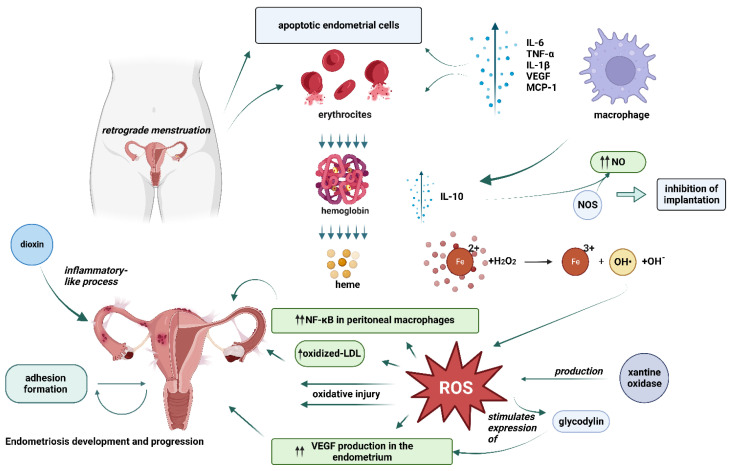
Potential mechanisms of oxidative stress related-endometriosis development. One of the many mechanisms is related to the activation of immune system caused by endometrial cells entering the peritoneal cavity during retrograde menstruation, which leads to excess reactive oxygen species (ROS) production by the activated macrophages, apoptotic endometrium cells, and erythrocytes. Iron, the by-product of hemolysis, then may act as a catalyst in the Fenton reaction further generating ROSs. Apart from direct oxidative injury to endometrial cells, ROSs can also increase the production of nuclear factor-kappa B (NF-κB) and vascular endothelial growth factor (VEGF) which play a role in the immune and inflammatory response in endometriotic cells and lead to an increase in levels of cytokines, chemokines, adhesion molecules, growth, and angiogenic factors all of which are related to the development and progression of endometriosis. (This figure was created with BioRender.com).

## Data Availability

Not applicable.

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
