# Peer review of "Oxidative Stress as a Potential Underlying Cause of Minimal and Mild Endometriosis-Related Infertility"

_ijms, 2023, doi:10.3390/ijms24043809_

Round 1

Reviewer 1 Report

The topic of the manuscript is of interest. The manuscript itself is not.

Many speculations, very few real data. A systemastic review would be needed.

Suggestions of antioxidant treatments that are not evidence-based is simply completely wrong.

If the authors would like to publish a meaningful review, then they should prepare tables with biomarker studies focused on oxidative damage, animal experiments analyzing the role of oxidative stress and clinical studies testing antioxdants.

Then they should discuss the observed consensus outcomes.

And of course, they should either focus on endometriosis OR on infertility. These are surely not synonyms. 

Author Response

Dear Sir/Madam,

Thank you very much for Your time, observations and for raising important questions.

In this review article we want to raise only hypotheses and look at the opposite point of view of this topic. That is why we have chosen a new controversial idea (topic) of a posibility that oxidative stress may be a potential underlying cause of minimal and mild endometriosis-related infertility. Endometriosis being one of the most common infertility-associated diseases is suggested to be one of the causes of increased oxidative stress. However, there is evidence that increased ROS production prompted by other factors could also be the cause of endometriosis development as well as of co-existing oxidative-stress-related infertility. While endometriosis is a benign diagnosis, an estimated 30-50% of women with endometriosis are infertile and, according to different sources, up to 90% of infertile women are diagnosed with endometriosis, this includes minimal and mild endometriosis which should not be an infertility cause. We wanted to speculate on the future and research possibilities regarding the effect of oxidative stress on development of endometriosis. We are only speculating about future treatment options when more research is done. We certainly do not suggest that all patients should require antioxidant therapy without further investigation, although some publications suggest this. We clarified this in our publication. Since this is a new idea, and there is a lack of literature on this topic, it is not yet possible to conduct a systematic review in order to clarify the raised question and not leave behind any bias. Additional research is needed, and we believe that the theory we put forward will be an inspiration for new studies that will allow the scientific community to clarify this topic more precisely. We are really sorry for misconception, we don't use endometriosis as a synonym for infertility, however, studies show that infertility of unknown origin is not uncommon in women whit minimal and mild endometriosis and these separate conditions (infertility and endometriosis) may be caused by oxidative stress.

Sincerely,
Authors

Reviewer 2 Report

Please rewrite 2nd sentence of abstract in proper English. (lines 12-14) Please provide reference for the statements in lines 88-93 Line 151 - Aging is spelled without an 'e' Line 158 - DOR is associated with diminished QUANTITY, but not Quality of the oocytes in most studies Following the DOR discussion, I would mention information linking endometriosis with DOR Line 205 - please change to "has a higher chance of survival"...

Author Response

Dear Sir or Madam,

Thank you very much for Your time reviewing our manuscript and for your very accurate observations. We have corrected our manuscript accordingly. Please see the attachment.

Sincerely,
Authors

Reviewer 3 Report

Dear Authors, thank you.

Author Response

Dear Sir or Madam,

Thank you very much for Your time reviewing our manuscript.

Sincerely,
Authors

Reviewer 4 Report

Dear

This research documented novel access to endometriosis as one of the most frequent causative factors of primary and secondary infertility. I think this research is adequate for publishing.

Sincerely

Prof Ivana Babovic MD.Ph.D

Faculty of Medicine, University of Belgrade Serbia

Author Response

Dear Prof. Ivana Babovic MD.Ph.D,

Thank you very much for Your time reviewing our manuscript.

Sincerely,
Authors

Round 2

Reviewer 1 Report

The manuscript type is "Review" - it should review the currently available evidence. The authors in their reply use the word "speculate" and "speculating" and that is the problem. The researchers do not need a review of speculations.

The authors are right "there is a lack of literature on this topic". How can You prepare a review if there is a lack of literature? The authors are encouraged to submit their results from experiments or clinical studies.

Otherwise, Medical Hypotheses would be a suitable journal, but even for that one needs backup from the literature.

Although the authors would likely not agree, such manuscripts/papers can be very dangerous as they are read not only by researchers and are often wrongly interpreted by patients who do not distinguish between evidence and speculation.